# Factor Analysis of Students' Knowledge Assessment Based on the Results of Online Entrance Testing in Mathematics to the University under the Conditions of COVID-19

Anastasiia Safonova [1,*] and Mikhail Guner [1,2]

1 Department of Information Systems, Saint Petersburg Electrotechnical University LETI, 197022 Saint Petersburg, Russia
2 Department of Artificial Intelligence Systems, Institute of Space and Information Technologies, Siberian Federal University, 660074 Krasnoyarsk, Russia
* Correspondence: safonova.nastya1@gmail.com

**Abstract:** After the COVID-19 pandemic emerged, many educational institutions switched to distance learning, which led to the problem of organizing entrance examinations for universities. When conducting entrance examinations online, applicants have many more opportunities for fraudulent activities (cheating on the exam and using outside help). This article is devoted to the factor analysis of the assessment of students' knowledge in mathematics enrolled in 2020 at the Altai State Technical University (Barnaul, Russia) based on the results of online entrance testing during the COVID-19 pandemic using an Internet browser and the webcams of applicants. The study used statistical methods, including groupings and comparisons. The analysis revealed that the spread of students' entrance ratings and their grades at the end of examination sessions in the first year increased by 2.7 times compared to previous years when entrance testing was held offline at the university. Such a large spread can be justified by 37% of the personal problems of applicants (difficulty adapting to the educational process at the university, lack of time, change of interests, etc.) and by 17% of a partial transition to distance learning in the current COVID-19 pandemic. There is a 46% imperfection rate of online testing as a mechanism for competitive selection of applicants. Despite the moderator's constant control over the progress of testing by monitoring the video from the cameras of applicants, some students probably used outside help in the exam. A correlation analysis showed that the propensity to such behavior is influenced by such characteristics as the age of the applicant, the level of previous education, knowledge of Russian for foreigners, degree of adaptation to life, and education in another country. In addition, the analysis of the correctness of answers to the tasks of the online test made it possible to identify several tasks that can additionally serve as a detector of independence when passing the test, called "traps".

**Keywords:** factor analysis; correlations; university education; distance technologies; online testing; competitive selection of students; higher education; COVID-19

## 1. Introduction

The COVID-19 pandemic has led many educational institutions around the world to switch to distance learning technologies. This process is called "emergency distance learning" and has caused many problems in the educational environment (Topuz et al., Schultz et al., Gamage et al., Lassoued et al., Saikat et al., and Reedy et al. [1–6]). One of these problems is the organization of entrance examinations to the university. When conducting entrance examinations online, applicants have many more opportunities for fraudulent activities (cheating on the exam and using outside help). The task of detection and suppression of such actions is the responsibility of the university admissions committee. The quality of the contingent of students enrolled in the university depends on the performance and safety indicators, the scientific indicators of the university. The relevance

of the work is due to the need to improve the effectiveness of online testing as a tool for the competitive selection of the most prepared applicants for admission, given the number of university students. The novelty of the study lies in the quantitative assessment of the effectiveness of online testing in comparison with classical applicant selection tools.

One of the problems is that the process of admission to the university requires entrance examinations, the purpose of which is to select the most prepared students. Robu et al. [7] point out the advantages of using new technologies in comparison with classical paper-based tests with the example of the Sim Entrance Exam (SEE), one of the online platforms for testing applicants. The advantages include the ability to analyze test results through the automated construction of various statistical graphs and the ability to scale and adapt to the requirements of any university. The architecture of such an online platform consists of three interrelated modules: the online learning environment, simulation of entrance testing, and administration. As a rule, the test questions are divided into several topics, covering the entire range of material for the entrance exam, with each question having a score depending on its difficulty. The simulation of testing can take place both in training mode and in the mode corresponding to the real conditions of the exam.

Moreover, the influence of socio-demographic and other factors on the result of the entrance exam was considered by Early et al. [8]. The authors conducted their study using the example of an entrance test for the Undergraduate Medicine and Health Sciences Admission Test (UMAT) in Australia and New Zealand. A significant proportion of all candidates retake the UMAT. An increased likelihood of retaking the UMAT two or more times was predicted by Pinheiro for young males who were non-English speakers and were from New Zealand; for Australian candidates, the likelihood of retaking was often determined by whether they were urban or rural residents.

As an example, the admission to public universities in Brazil depends on the scores obtained by high school students in the Exame Nacional do Ensino Médio (ENEM) (Silveira et al. [9]). ENEM is essentially a kind of analogue of the Unified State Exam (USE) in the Russian Federation (RF). ENEM consists of four tests in language, mathematics, science, and humanities. Other articles compare the performance of students at universities and colleges with their grades based on the results of the entrance exam and also evaluate the factors influencing student performance in certain subjects, in particular in physics. Studies by scientists have shown that the gender of the student and the sector of the secondary school (federal, regional, or private schools) are very significant here (Pinheiro et al., Barroso et al., da Silva et al. [10–12]).

German researchers asked whether exam scores are predetermined by students' previous performance and personal characteristics, or can lagging students catch up (Schwerter et al. [13])? During the semester, 281 undergraduate business students were offered additional practice in a university math e-learning course, which resulted in some students receiving extra credits for doing so. The voluntary practice was found to have a statistically significant positive effect on exam scores. The effect is especially noticeable for students who are lagging in the program, from whom one would otherwise expect poor results on the exam.

A very interesting study is that by Gawlak et al., in which they compare the skills of applicants and graduates of the Faculty of Architecture of one of the universities in Poland [14]. Due to the COVID-19 pandemic, the 2020 entrance exam excluded an assessment of candidates' drawing skills. The aim of the study was to find a correlation between the quality of candidates accepted into an undergraduate program, assessed on the basis of their drawing skills demonstrated in the entrance examination, and the learning outcomes of graduates. Comparative analyses have shown that excellent drawing skills at the beginning of training do not guarantee better learning outcomes at the end of training. The authors believe that a candidate's portfolio of work may well replace the entrance exam and become an effective tool for the competitive selection of applicants. In another study, UK researchers conducted a comparative study of the scores obtained in the distance and campus forms of passing the exam at school (Jaap et al. [15]). The average score of 4th

grade students on the results of the online exam was higher than that of those who took the exam in person.

Another paper has investigated the impact of the COVID-19 pandemic on student cheating in online exams at US universities (Abdelrahim et al. [16]). The results of the study showed a significant positive relationship between COVID-19 quarantine and anxiety, COVID-19 quarantine, and stress. Anxiety and stress, in turn, have a strong influence on student behavior, and students begin to cheat more on online exams.

Conducting examinations with sufficient reliability and efficiency has become one of the most important and complex issues in higher education (Noorbehbahani et al., Gorgani and Shabani [17,18]). During the COVID-19 pandemic, many universities have implemented online exam proctoring technologies to track student fraud. Although it looks like a natural and effective solution for a fair assessment of student performance in online learning. Researchers in South Korea argue that proctoring technologies have a negative impact on the relationship between students, students and the teacher, and, as a result, on learning outcomes (Lee and Fanguy [19]). However, Fask et al. [20] assessed the difference in performance between students taking a traditional exam with proctoring and those taking an online exam without proctoring. They emphasize the importance of the influence of the online testing environment, which characterizes the ability of students to cheat.

However, we have not come across a work that has analyzed the results of using online testing as a mechanism for the competitive selection of students for their subsequent enrollment in higher educational institutions.

The purpose of this study was to build a factor analysis of the assessment of the knowledge of students in mathematics enrolled in 2020 at the Altai State Technical University named after I.I. Polzunova (AltSTU) based on the results of the entrance testing conducted during the COVID-19 pandemic in an online format using an Internet browser and a webcam of applicants.

The main contributions of this paper can be listed as follows:

- As far as we know, this is the first work addressing the problem of quantifying the objectivity of the university entrance exam results in an online format.
- We collected a new dataset of students' performance in mathematics based on online entrance tests and exam sessions. The dataset also contains socio-demographic indicators, including gender, age, citizenship, level of previous education, etc.
- We conducted a factor analysis of students' knowledge assessments based on the results of online entrance testing to the university under the conditions of COVID-19.
- We conducted a comparative analysis of students' grades in mathematics before enrolling at the university (according to the results of the university entrance exam and the USE, in online and offline formats) and after enrolling at the university (according to the results of examination sessions in the first year).

The rest of the paper is organized as follows. The related works are provided in Section 2. Section 3 describes the characteristics of the data sample, the methods used, and the software and hardware. The obtained results are presented in Section 4, which includes an analysis of the distribution of a data sample, factors of influence on the dispersion of students' grades in mathematics based on the results of the entrance examination, the basis of the results of the examination sessions, and factors of influence on the assessment of students' knowledge in mathematics upon admission to the university based on the results of online testing and subsequent training. Discussion and conclusions are provided in Section 5.

## 2. Related Works

In this section, we consider related works devoted to the problems of using online technologies in higher education.

Malaysian researchers identified four main components of online learning: the quality of the developed course, communication in the course, time management, and technical

competence of online learning participants (Adi Badiozaman and Segar [21]). Another example is that Vietnamese scholars have focused on the diversity of interaction models (teacher–student, student–student, and student–content) in online learning during the pandemic. The results of their study showed that most teachers deployed activities for two main types of interaction: teacher–student and student–content, but not for student–student interaction (Le et al. [22]). It is quite possible that it is the lack of communication between students in online practices that determines the sometimes observed decline in the quality of education received in the online learning format.

The Teaching in the Post-COVID-19 Era book describes the practical educational technology solutions implemented during the COVID-19 situation and the implications for the future of the education system (Fayed [23]). The projects of specific programs and innovations in online learning curricula that meet the needs of international students in higher education are presented in detail. Majola and Mudau [24] describe, in detail, the problems of students and the experiences of teachers at universities in South Africa in terms of distance learning during the COVID-19 pandemic. The main problems for students were the correct distribution of their time, connection to the Internet, and access to digital resources, including due to power outages and a strong increase in the load on network communication channels. Teachers were forced to revise the methods of assessing students' knowledge, conduct online exams for a while, pay more attention to safety during exams, and identify and prevent violations. At the Comenius University in Slovakia, teachers asked students to record videos in which they described the progress of their experiments to further assess students' knowledge of physics (Šromeková [25]).

Obviously, the progress of students and, consequently, the safety of the composition of students depends on the area of their interests. Ural Federal University (RF) researchers proposed an approach to the analysis of students' educational interests based on data from social networks (Komotskiy et al. [26]). They collected data on 1379 students studying at three institutes. These students were grouped into clusters based on their interests in the Russian social network Vkontakte (VK). These clusters were then compared with the institutions where the students study. Thus, this approach allowed researchers to successfully divide students into those who are interested in computer science, humanities, and social sciences. As a result, information about the characteristic interests of students can be used for the preliminary selection and invitation of university applicants and schoolchildren to the relevant educational areas. In addition, social media data, combined with other data sources, will allow better prediction of student achievement. However, another study found qualitative changes in university education caused by the pandemic, from the point of view of university authorities (Navickiene et al. [27]).

In their paper, Guncaga et al. [28] presented the results of a survey of students about online learning at universities in Slovakia, the Czech Republic, and Kazakhstan during the COVID-19 pandemic. Most of the difficulties, according to students, are related to their social status, lack of social contacts, technical problems with connecting to the Internet, organizing lectures, and taking exams. A recent study was conducted on the psychological record and academic experiences of university students in Australia (Dodd et al. [29]). Multivariable regression models showed female gender, low subjective social status, a negative overall learning experience, or reporting COVID-19 as having a huge impact on study were associated with lower wellbeing in the first few months of the pandemic. In the work of Etajuri et al. [30], the impact of the pandemic on the physical and mental health of students was studied. They surveyed 150 dental students at a Malaysian university. About 66% of students felt comfortable adapting to new technologies, and 85.7% were concerned about the quality of online learning. Almost all students, 98.6%, expressed doubts about their ability to pass qualifying exams and complete their studies on time, and only 49.7% agreed that clinical experience was effectively gained through online learning. Another interesting article analyzes why students turn their webcams on or off during online classes (Gherheș et al. [31]). The results highlighted the fact that more than half of the students participating in the study reported that they do not agree to keep their webcams on during

online classes, the main reasons being anxiety/fear of being exposed/shame/shyness, the desire to ensure privacy of the home/personal space, and the possibility that other people might walk into the background. Moreover, the University of Ljubljana revealed a difference in the attitude of students and employees towards working from home and online education during the COVID-19 pandemic (Varineja Drašler et al. [32]).

Canadian researchers also conducted a survey (Chen et al. [33]). The results of their study showed that, with the effective use of online tools, distance learning can replicate the key components of educational content delivery. However, educators and students do not want face-to-face learning to disappear and consider it right to flexibly combine face-to-face and online learning opportunities. German researchers statistically proved that online tests can be effectively used not only to diagnose learning outcomes but also to improve the learning process in general (Wittrin et al. [34]). In their opinion, online tests have a positive effect on student motivation, and the authors emphasize the importance of systematic digital testing in learning management systems. Other work conducted by Spanish researchers concerns the impact of COVID-19 on university staff and students from Iberoamerica: online learning and teaching experiences (Jojoa et al. [35]).

In the works we reviewed, the advantages of online technologies, the problems of their implementation in education, and the impact of distance teaching on the quality of knowledge received were often noted. However, the most similar work to ours was conducted by Comas-Forgas et al., who used the emerging research method of search engine data analysis to investigate the extent of requests for exam cheating information in Spain in the time period surrounding adjustments for the pandemic [36]. However, as far as we know, our work is the first to address the problem of quantifying the objectivity of the university entrance exam results in an online format using statistical methods, including groupings and comparisons.

## 3. Materials and Methods

### 3.1. Characteristics of the Data Sample

The initial data in this work were the data of students on admission to AltSTU and the subsequent education of 553 students enrolled in 2020 based on the results of an online entrance test conducted during the COVID-19 pandemic. The sample of students is limited only to the students who passed an entrance exam in mathematics and subsequently studied mathematical disciplines.

There were 402 boys and 151 girls in the total number of subjects.

The distribution by age composition was as follows: 62 people under the age of 18 as of 1 November 2020, 270 people aged 18–21, 86 people aged 22–25, 46 people aged 26–30, 48 people aged 31–35, 22 people aged 36–40, and 19 people over 41 years old. The histogram of the distribution of the number of students who passed online entrance tests in mathematics and enrolled in the university in 2020 is shown in Figure 1.

The division by citizenship was as follows: 395 citizens of the RF, 135 citizens of Kazakhstan, 20 citizens of Tajikistan, and three citizens of other states. On a territorial basis, 361 people were from urban areas and 192 people were from the countryside.

300 people were enrolled in budget places, including 287 on a general basis and 13 on preferential and targeted quotas. A total of 253 people were enrolled in paid places of study. By forms of education, 225 people were enrolled in full-time education, and 328 people were enrolled in full-time and part-time education.

Applicants were allotted 180 min to pass the exam in mathematics. Moreover, the average time spent on passing the exam was recorded—120 min, the minimum—19 min, and the maximum—180 min.

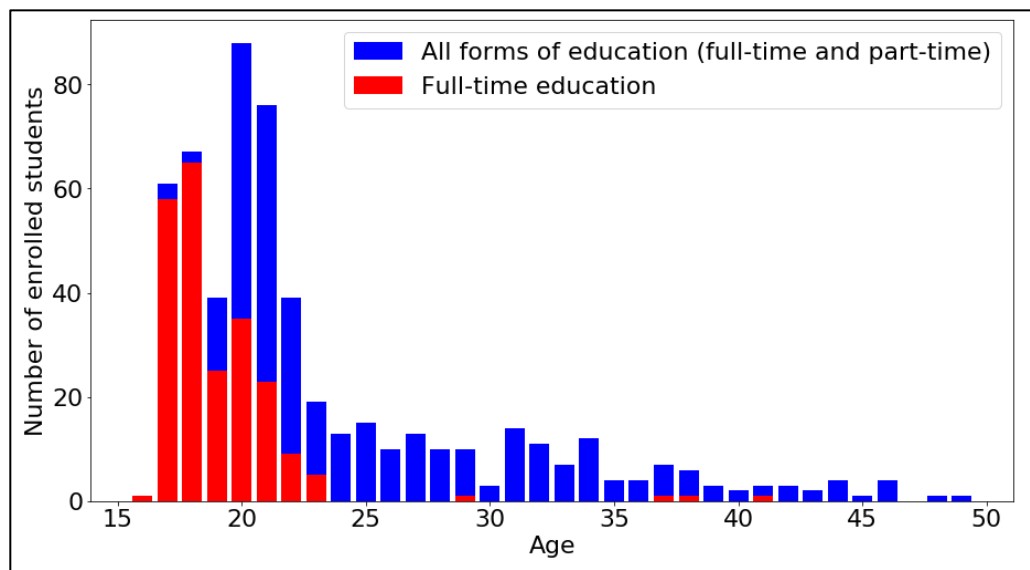

**Figure 1.** Histogram of the distribution of the number of students who passed online entrance tests in mathematics and were enrolled in AltSTU, 2020.

The minimum positive test result was 39 points, the maximum test result was 100 points, and the average value was 81.6 points. As you can see from Figure 2, most applicants completed the test in 130–135 min or 175–180 min. The most successful students completed the online math test in 105–140 min.

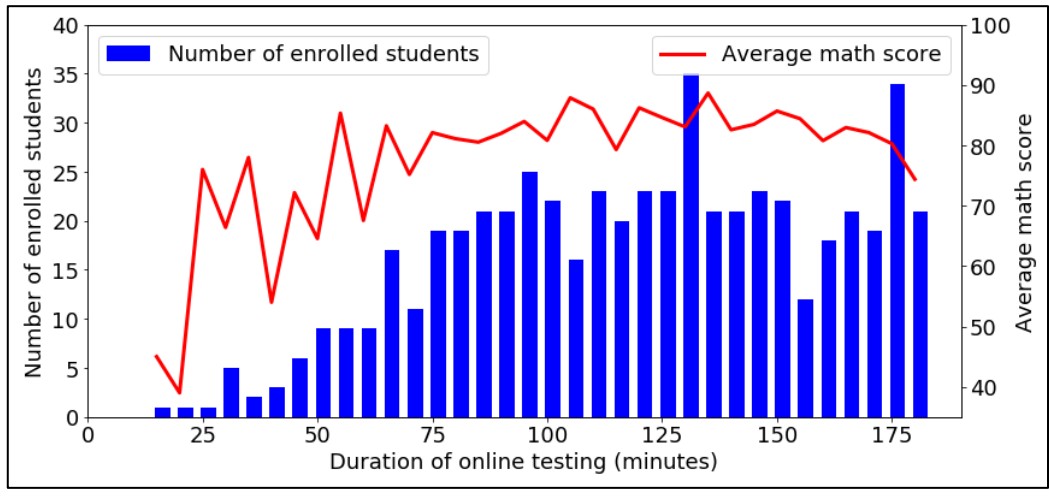

**Figure 2.** A histogram of the distribution of the duration of passing an online test in mathematics by applicants (in minutes) and a graph of the dependence of the test result on the duration of passing.

Baseline data on admission and subsequent studies of 553 students enrolled in the university in 2020 based on the results of the online entrance test in mathematics (for the example of two students) are given in Appendix A.

Additionally, in order to compare the results of the study, data were taken on the input ratings and ratings based on the results of examination sessions of students enrolled based on the results of the USE, as well as similar data on students enrolled three years earlier (Table 1).

**Table 1.** Distribution of the number of students enrolled in AltSTU based on the results of university testing and the results of the USE in mathematics (from 2017 to 2020).

| Year | According to the University Testing | According to the USE |
|---|---|---|
| 2017 | 641 | 1271 |
| 2018 | 504 | 1195 |
| 2019 | 576 | 1000 |
| 2020 | 553 | 803 |
| Total | 2274 | 4269 |

In addition, it should be noted that between 2017 and 2019, all entrance examinations (university testing and the USE), training, and examination sessions were held exclusively offline. In 2020, the USE was still conducted offline, university entrance testing was conducted online, and full-time freshman training and examination sessions were conducted, for the first time, using remote technologies.

*3.2. Methods*

In our study, statistical methods are actively used, namely, finding the minimum (1), maximum (2), and average (3) values, dispersion (4), grouping methods (5), methods of comparison, and comparisons (Scheffé [37]):

$$\min(n1,\ n2,\ n3,\ \ldots), \tag{1}$$

where $n1$, $n2$, and $n3$ are one or more items to compare.

$$\max(n1,\ n2,\ n3,\ \ldots), \tag{2}$$

$$\overline{x} = \frac{\sum_{i=1}^{n} x_i}{n}, \tag{3}$$

where $x_i$ is $i$-th element of the array, and $n$ is the number of values in the array.

$$var(X) = \frac{\sum_{i=1}^{n} (x_i - \mu)^2}{n}, \tag{4}$$

where $\mu$ is arithmetic mean calculated by the Formula (3).

$$(X, Y) => (unique\ X,\ \forall\ (x_k \in unique\ X)\ func(\{y_i | x_i = x_k\})), \tag{5}$$

where $(X, Y)$ is the set of pairs of values $(x_i, y_i)$, *unique X* is a set of unique $x_i$, *func* is one of the functions: number of values, sum of values, min (1), max (2), average (3) value.

To assess the interdependencies between the indicators, linear correlation coefficients were used (6) (Leischner [38]).

$$r_{xy} = \frac{\sum_{i=1}^{n} (x_i - \overline{x}) * (y_i - \overline{y})}{\sum_{i=1}^{n} (x_i - \overline{x})^2 * \sum_{i=1}^{n} (y_i - \overline{y})^2}, \tag{6}$$

where $n$ is the number of rows in the sample, $x_i$ is the $i$-th value of the $x$ parameter, $y_i$—is the $i$-th value of the parameter $y$, $\overline{x}$, and $\overline{y}$ are the average values of the parameters $x$ and $y$ over all examples. The linear correlation coefficient $r_{xy}$ between $x$ and $y$ can take values from $-1$ to $+1$.

*3.3. Software and Hardware*

Online testing of students took place in the author's system, developed on the basis of AltSTU. The system was developed in the PHP programming language using the Bootstrap framework, JavaScript scripting language, AJAX technology, and CSS style language. The

online testing system is available on the Internet. However, registration in the system is closed and available only to applicants who have submitted an application.

An obligatory requirement for passing the entrance test in the online format was the broadcast of the exam to applicants through a webcam connected to their personal computer or through the front camera on their smartphone. Online tests were monitored in real-time by moderators from the university (based on one moderator per virtual office with no more than 24 test takers).

## 4. Results

### 4.1. Analysis of the Distribution of a Data Sample

Analysis of the distribution of the data sample was carried out in several sections. First, the results of university online testing in mathematics of the enrolled students were compared with the results of examination sessions. As can be seen in Figure 3a, the distribution densities of the scores of 553 students in mathematics based on the results of introductory online testing and examination sessions while studying at the university do not largely coincide. This indicates the problem of objectivity in assessing the knowledge of applicants through online testing.

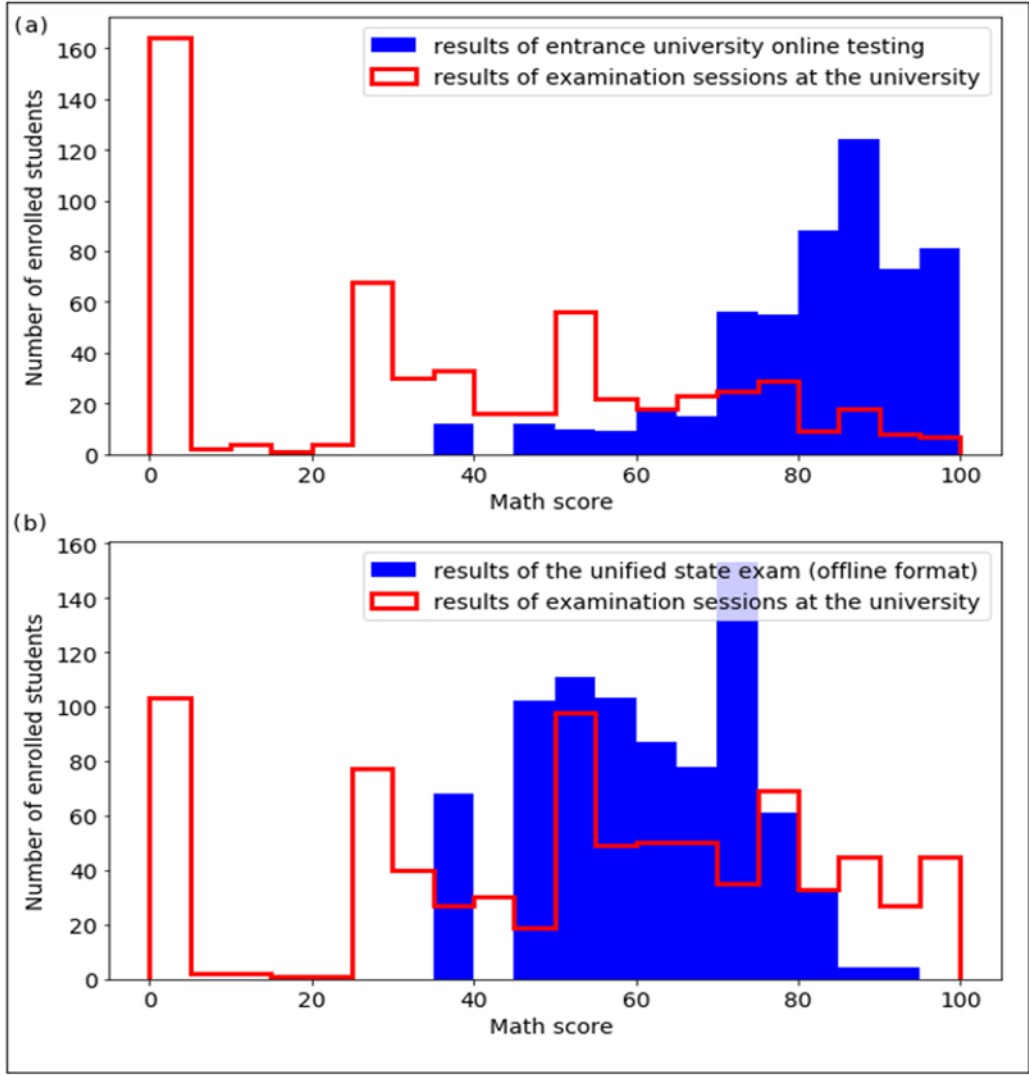

**Figure 3.** Histograms of the distribution of grades in mathematics based on the results of online entrance testing (**a**) and the USE in offline format in comparison with the results of examination sessions (**b**).

For comparison, additional data were taken on 803 students enrolled in 2020 based on the results of the USE. Based on these data, a histogram of the distribution of marks in mathematics was also constructed (Figure 3b). The distribution densities of the estimates also do not coincide.

The average score in mathematics of 553 students enrolled on the basis of university online testing during the course of university studies was 35.94 points. The average score in mathematics of all 1356 students at the end of the examination sessions is 45.54 points.

It should be noted that the scaling systems for competitive selection to the university and during examination sessions are different: a positive grade in mathematics, required for admission to the university in 2020, was 39 points, and when studying at the university itself, it was 25 points.

In addition, according to the procedure for admission to universities in the RF in 2020, only certain categories had the right to enter based on the results of university online testing and not the USE, which includes:

- Persons who have a document of secondary vocational education (SVE) or a document of higher education (HE).
- Foreign citizens.
- Citizens of the RF who in 2020 received a document on secondary general education in a foreign organization and did not pass the USE.
- Disabled people and children with disabilities.

Data on the number of students enrolled in the university in 2020 for various forms of education based on the results of the entrance exam in mathematics, in various sections, are shown in Tables 2 and 3.

**Table 2.** Data on the distribution of the number of 1356 students enrolled in the university in 2020 based on the results of the entrance exam in mathematics, by form of education.

| Form of Education | University Online Testing | USE |
|---|---|---|
| Full-time | 225 (16.6%) | 761 (56.1%) |
| Part-time | 328 (24.2%) | 42 (3.1%) |
| Total | 553 (40.8%) | 803 (59.2%) |

**Table 3.** Detailing the number of students enrolled in the university in 2020 based on the results of the entrance exam in mathematics, by forms of study, categories of citizenship, and type of document on education.

| Form of Education | Student Category, Type of Education Document | University Online Testing | USE |
|---|---|---|---|
| Full-time | Foreign citizens | 140 (62%) | 0 (0%) |
| | RF citizens: | 85 (38%) | 761 (100%) |
| | 1. Certificate of secondary general education | 4 [1] | 753 |
| | 2. Vocational diploma | 81 | 8 |
| Part-time | Foreign citizens | 18 (5%) | 0 (0%) |
| | RF citizens: | 310 (95%) | 42 (100%) |
| | 1. Certificate of secondary general education | 2 [1] | 38 |
| | 2. Vocational diploma | 278 | 4 |
| | 3. HE diploma (bachelor's, specialist's, or master's degree) | 30 | 0 |

[1] Citizens of the RF who received a certificate of secondary general education in a foreign organization (or disabled people, disabled children).

As can be seen from Table 2, the number of students enrolled in full-time education based on the results of the USE is 3.4 times higher than the number of students enrolled in full-time education based on the results of university online testing (761 to 225).

At the same time, among those enrolled in part-time and part-time forms of education, the relationship is reversed: according to the results of university online testing, 7.8 times more students were enrolled than according to the results of the USE (328 to 42).

It should be noted that among those enrolled on the basis of online university testing for full-time education, the majority (62%) are foreign citizens, and for full-time and part-time education, the majority (95%) are citizens of the RF, who usually have a diploma of secondary vocational education or higher education.

The USE was taken mainly by citizens of the RF with a certificate of secondary general education (graduates of secondary general education schools). Passing the exam for graduates of secondary schools in the RF is necessary to obtain a certificate. In addition, according to the results of the exam, you can enter all universities in the country.

It is also important that the USE in 2020, as always, was conducted offline, while the university testing of the same year was transferred to an online format due to the COVID-19 pandemic.

In this regard, it was decided to consider additional data on the admission and subsequent education of students at the university enrolled in the period from 2017 to 2019. based on the results of university testing and the USE conducted offline (only in 2020 was university testing conducted online). The deviation between the average score of students in mathematics at admission and at the end of examination sessions in 2020 increased sharply to 45.6 points (Table 4).

**Table 4.** Baseline data on students enrolled based on the results of university testing in the period from 2017 to 2020.

| Admission Year | NUMBER OF ENROLLED | Average Score at Admission | Average Score at the End of Examination Sessions | Deviation (Compared to 2020) |
| --- | --- | --- | --- | --- |
| 2017 | 641 | 61.13 | 38.73 | −22.40 (23.22) |
| 2018 | 504 | 51.95 | 36.20 | −15.75 (29.87) |
| 2019 | 576 | 52.97 | 40.47 | −12.50 (33.12) |
| 2020 | 553 | 81.56 | 35.94 | −45.62 |

In the context of the COVID-19 pandemic, student education in 2020–2021 was carried out mainly with the help of distance technologies (for full-time education—for the first time). To assess the influence of this factor, the absolute value of deviations in 2017–2019 was studied. between the average scores of students in mathematics at admission and at the end of examination sessions in comparison with the deviation in 2020 (Table 5).

In this study, the verification of the results of the entrance university testing in mathematics as a tool for assessing the real knowledge of the applicant was carried out by comparing the entrance ratings of students with the results of subsequent studies at the university.

**Table 5.** Data on deviations between the average scores of students in mathematics at admission and at the end of examination sessions for 2017–2019 compared to deviations in 2020.

| Admission Year | Deviation [1] (According to Those Enrolled Based on the Results of the USE) [2] | Deviation [1] (According to Enrolled Students Based on the Entrance University Testing) [3] |
|---|---|---|
| 2017 | 10.1 | 23.2 |
| 2018 | 7.7 | 29.9 |
| 2019 | 5.3 | 33.1 |

[1] Deviation between the average score of students in mathematics at admission and at the end of examination sessions in comparison with such a deviation in 2020. [2] In all years (from 2017 to 2020), the USE was conducted in an offline format. The number of studied students enrolled based on the results of the USE in mathematics, by year of admission: 803 in 2020, 1000 in 2019, 1195 in 2018, and 1271 in 2017. [3] Entrance university testing was conducted offline in 2017, 2018, 2019, and online in 2020.

*4.2. Factors of Influence on the Dispersion of Students' Grades in Mathematics Based on the Results of the Entrance Examination and the Basis of the Results of Examination Sessions*

In this study, the spread of scores in mathematics based on the results of the entrance online testing in 2020 and examination sessions was 45.6 points (on a 100-point scale). Three years earlier, this spread of marks ranged from 12.5 to 22.4 points (average—16.9), which is 2.7 times less than in 2020 (Figure 4).

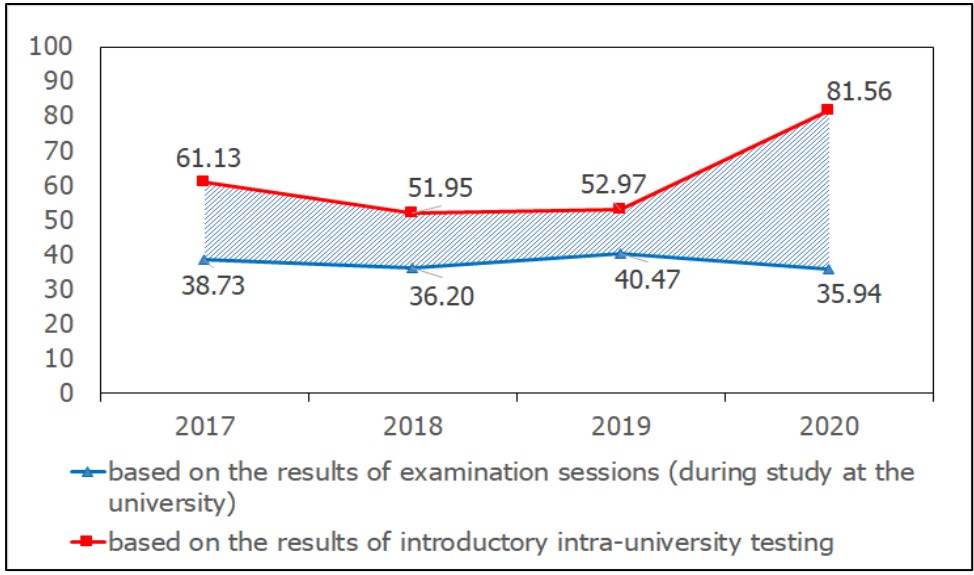

**Figure 4.** The spread of marks in mathematics is based on the results of entrance university testing and on the basis of examination sessions.

Thus, the large spread of grades in mathematics based on the results of university online testing and the results of examination sessions can be explained by the peculiarities of scaling, personal problems of applicants (difficulty adapting to the educational process at the university, lack of time, change of interests) by only 37 percent (in points: 16.9 out of 45.6). The remaining 63 percent of the spread of marks (in points: 28.7 out of 45.6) can be explained by two main reasons: the distance format of the entrance exam and the distance (partly distance) format of the first year of study.

Figure 5 shows a diagram of the dynamics of the scatter of deviations in estimates by year of receipt compared to 2020 based on the data from Table 5.

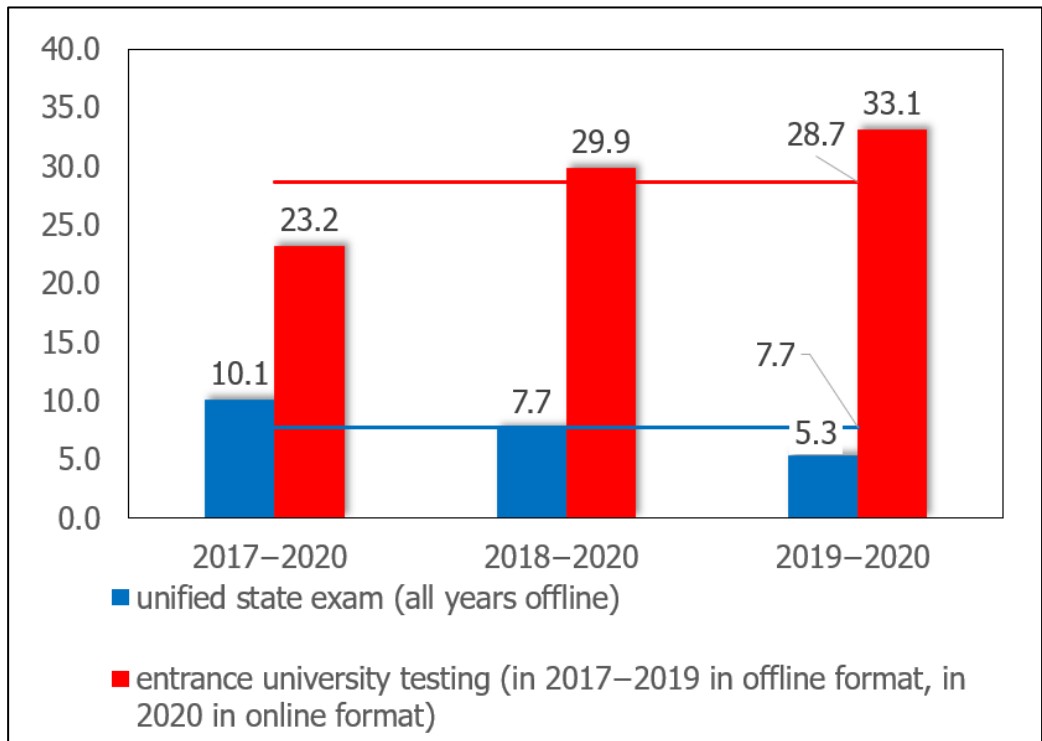

**Figure 5.** The dynamics of dispersion of average grades in mathematics at university entrance exams and based on the results of examination sessions for 2017–2019 compared to 2020.

As can be seen from Figure 5, the spread of scores in mathematics for the online entrance tests in 2020 and the results of examination sessions, compared with the past three years, when the entrance tests were conducted offline, was 28.7 points. At the same time, a similar spread of grades for applicants based on the results of the USE, which all these years took place exclusively in the offline format, amounted to 7.7 points.

Therefore, 17 percent (in points: 7.7 out of 45.6) of the deviations in the marks of applicants in mathematics in 2020 in the results of online entrance tests and examination sessions can be explained by the distance (partly distance) learning format in the 1st year. The online format of the entrance exam is the reason for the deviation in grades, which accounts for 46 percent, respectively (in points: 21 out of 45.6).

*4.3. Factors of Influence on the Assessment of Students' Knowledge in Mathematics upon Admission to the University Based on the Results of Online Testing and Subsequent Training*

The subsection presents factors of influence on the assessment of students' knowledge in mathematics upon admission to the university based on the results of online testing and subsequent training. The distribution quadrant of the 553 students enrolled based on the results of the online entrance test in mathematics is presented in Table 6.

**Table 6.** Distribution quadrant of 553 students enrolled on the basis of the results of the entrance university online testing in mathematics.

|  |  | Examination Session | |
|---|---|---|---|
|  |  | **High Math Score** | **Low Math Score** |
| **Entrance university testing** | **High math score** | 161 (29%) | 172 (31%) |
|  | **Low math score** | 67 (12%) | 153 (28%) |

The high score in mathematics at admission was higher than the average (81.6 points), according to the passing results of the university's online testing of 553 students. The high

score in mathematics when studying at the university was above average (45.54 points) according to the results of passing the examination sessions of all 1356 students, including 553 students enrolled based on the results of university online testing and 803 students enrolled based on the results of the USE.

As can be seen from Figure 6, only 12% of students with initially low input ratings improved it, while 31% of students with initially high input ratings worsened it. This once again emphasizes the need to improve the mechanism for the competitive selection of applicants.

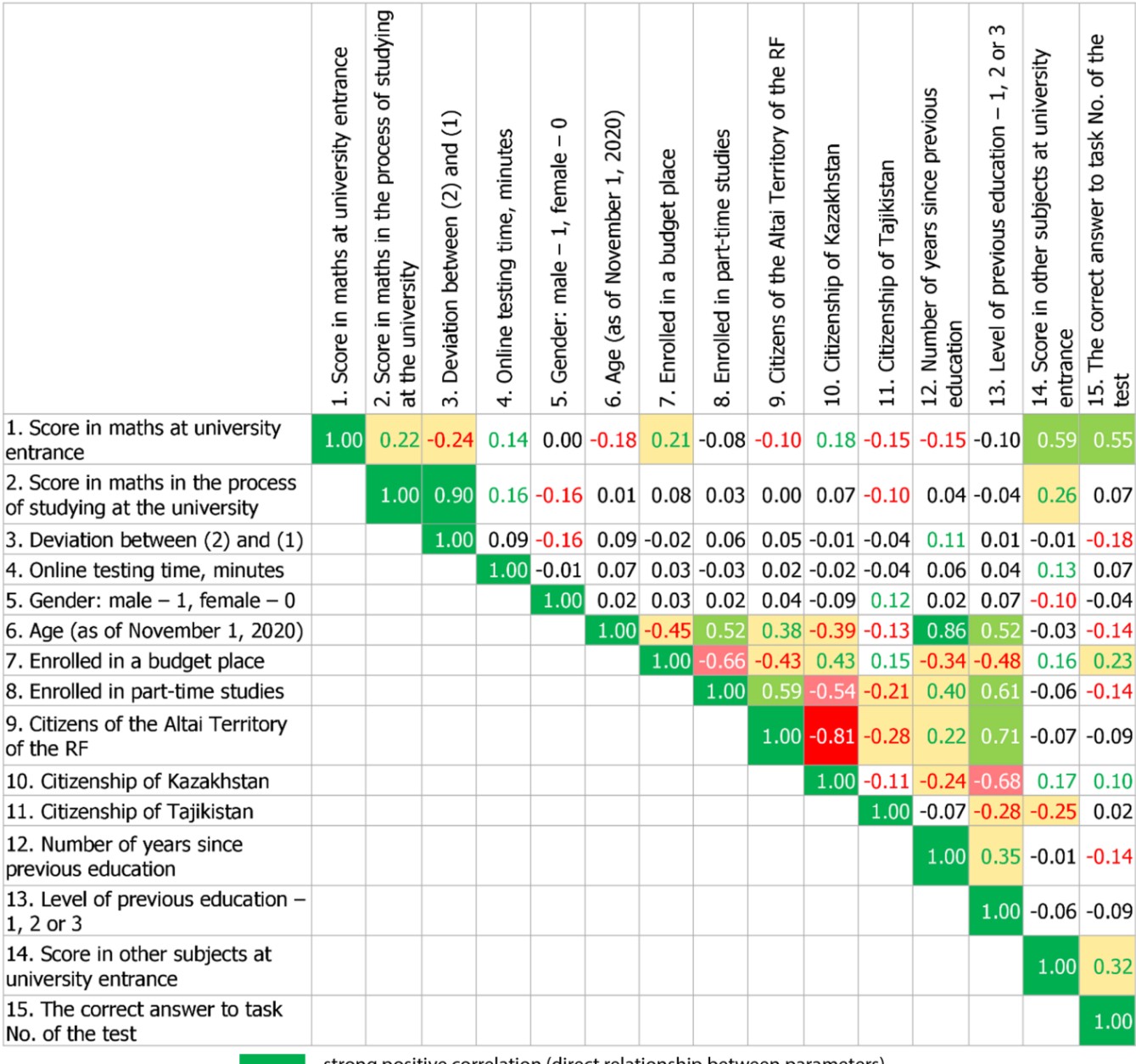

**Figure 6.** Matrix of linear correlation coefficients between the 15 main indicators in the model for assessing the level of knowledge of applicants and students in mathematics.

Let us turn to the initial data on the admission and subsequent education of 553 students (a fragment of the initial data sample is given in Appendix A). To determine the factors influencing the assessment of students' knowledge in mathematics upon admission

to the university based on the results of online testing, the assessment during subsequent training, and the deviation in these assessments, we will find linear correlation coefficients. Correlation coefficient values greater than 0.9 indicate the presence of a strong direct relationship between the parameters; less than 0.9 indicates the presence of a strong inverse relationship, and about 0 indicates the absence of linear relationships. The correlation matrix for 15 main indicators is presented in Figure 6.

The correlation between math scores at university entrance and at university is positive, but only 0.22. This means that the mechanism of competitive selection of students for enrollment based on the results of university online testing does not make it possible to accurately predict student performance in the future.

The deviation in the scores of students before and after enrollment is mainly explained by the value of the grades based on the results of examination sessions (correlation 0.9), which indicates a low dispersion in grades in mathematics upon admission and insufficient differentiation in the level of knowledge of applicants.

A long time to complete online testing often indicates the diligence of the applicant and has a positive effect on grades (correlations of 0.14 and 0.16).

As shown in Figure 7, better-trained students do perform better on university admissions online math tests in university exam sessions, but as entrance rankings increase, so too does the variance in students' pre- and post-matriculation math scores.

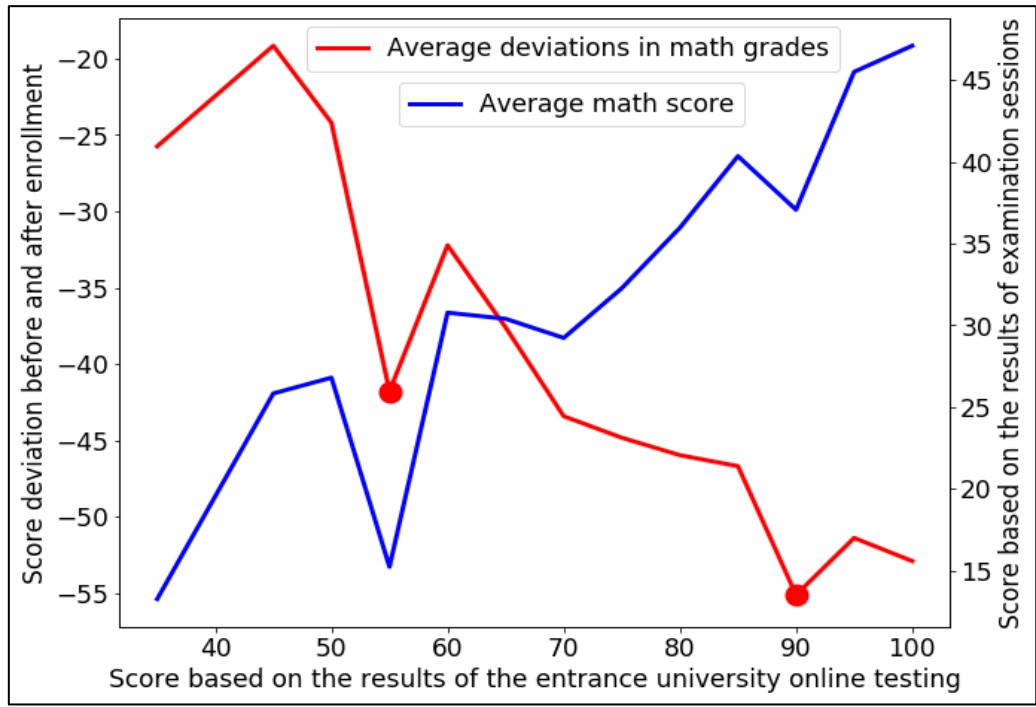

**Figure 7.** Graph of the average marks of students in mathematics when studying at the university based on the results of examination sessions and deviations in grades in mathematics before and after enrollment from the average grades upon admission based on the results of university online testing in 2020.

Two ranges of input rating stand out on the graph: 55–60 points and 90–95 points. Perhaps most applicants who took advantage of outside help in the online entrance test in mathematics scored this number of points. The goal of the first category of such applicants was to obtain a minimum positive assessment sufficient for enrollment in non-budgetary places or for enrollment in budgetary places in non-prestigious areas of training and specialties. The goal of the second category of such applicants is to achieve the highest possible score necessary to win the competition for budget places in prestigious areas of training and specialties.

The gender of the applicant does not affect his entrance rating in mathematics in any way; however, when studying at the university itself, female students show the best results in mathematics (correlation $-0.16$), and usually female students have higher input ratings in other subjects (for example, Russian language) (correlation $-0.1$).

The age of the applicant and the number of years since receiving the previous education negatively affect the ranking in mathematics at admission (correlations $-0.18$ and $-0.15$), but the situation improves somewhat during study at the university itself; the correlations of these two parameters with the deviation in students' scores before and after admission are 0.09 and 0.11. At the same time, most older students are citizens of the RF from the region where the university is located (correlation 0.38), and they are enrolled in distance learning (correlation 0.52) in paid places (the correlation with the "enrolled on the budget place" parameter is negative and amounts to $-0.45$). These applicants are admitted to university testing because they have a higher level of previous education (secondary vocational or higher education) (correlation 0.52).

Among foreign citizens, the best results at admission were shown by citizens of Kazakhstan (correlation 0.18), but after enrollment, the performance indicator deteriorates somewhat. At the same time, among enrolled students from Kazakhstan, women predominate (correlation $-0.09$), from Tajikistan men predominate (correlation 0.12). As a rule, young foreign citizens enter the university (correlations with age $-0.39$ and $-0.13$), and people from Kazakhstan and Tajikistan usually use the right to receive higher education in the RF, studying in state-funded places (correlations 0.43 and 0.15). Citizens of Kazakhstan and Tajikistan are usually full-time students (correlation with the parameter "enrolled in part-time education" is negative and amounts to $-0.54$ and $-0.21$).

All of the above may indicate that socio-demographic factors often determine the applicant's propensity to cheat and use outside help during the exam.

A direct relationship was also found between scores in mathematics and scores in other subjects when entering university (correlations 0.59 and 0.26). The analysis of the correctness of answers to tasks in the online test made it possible to identify several tasks, the success of which most affects the model for assessing students' knowledge in mathematics.

Of particular interest in our case is the parameter "correct answer to task No. of the test". The correlations between this parameter and the input rating in mathematics and the parameter "enrolled in a budget place" are 0.55 and 0.23, respectively. The relationship between the correctness of the answer to the item No. test and the number of years since receiving previous education is inverse (correlation $-0.14$). This suggests that not all applicants complete it (only 274 out of 553 students gave the correct answer), the task is difficult and is located at the end of the test. At the same time, the correlation between the correctness of the answer to the task No. of the test and the deviation in the students' scores in mathematics before and after enrollment is negative and amounts to $-0.18$. Thus, it is possible to put forward the hypothesis that applicants who gave the correct answer to the task No. could use qualified outside help.

As shown in Figure 8, students who received a high score (above 75–80 points) on the university online math test and correctly answered item No. of the test, perform worse during their university studies compared to those who answered the task No. of the test.

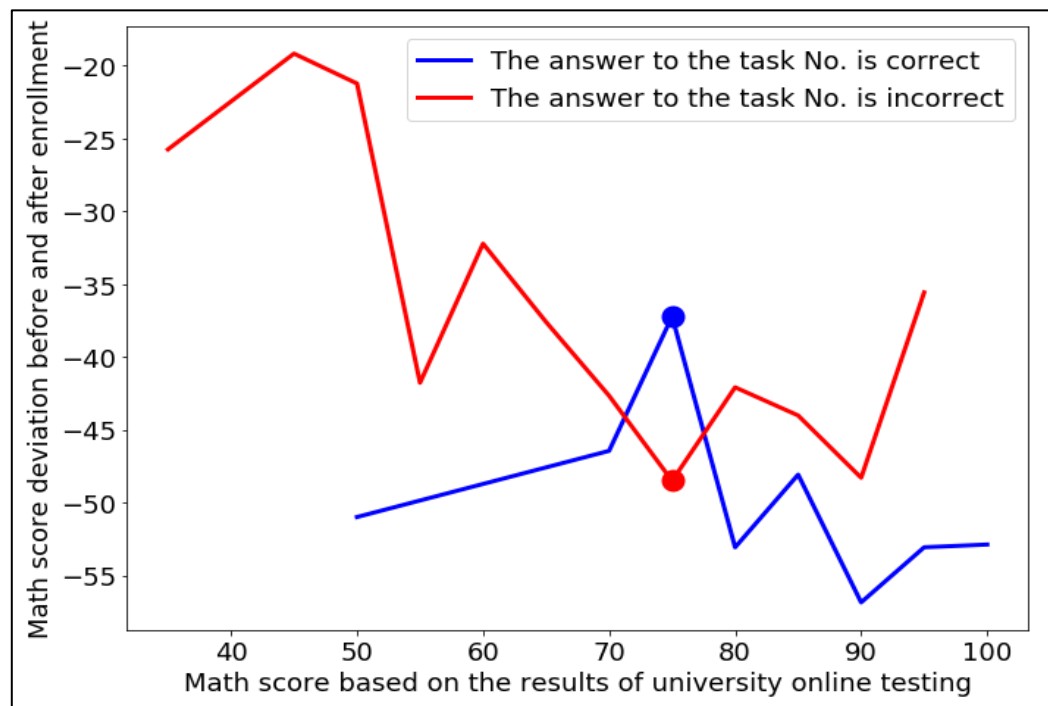

**Figure 8.** Dynamics of changes in deviations in students' grades in mathematics before and after enrollment, depending on the average grades upon admission based on the results of university online testing in 2020 and the correctness of the answer to the example No. task.

According to the results of the study, it should be concluded that the inclusion of "trap tasks" in online tests allows for assessing the real knowledge of applicants and drawing attention to the incoming from the moderator.

## 5. Discussion and Conclusions

The online format of conducting entrance testing for the university causes certain difficulties when trying to objectively assess the knowledge of applicants. In this study, the spread of grades in mathematics at AltSTU based on the results of the entrance online testing in 2020 and examination sessions was 45.6 points (on a 100-point scale). Three years earlier, this spread of estimates ranged from 12.5 to 22.4 (average—16.9). In other words, the deviation can be explained by scaling features and personal problems of applicants (difficulty of adapting to the educational process at the university, lack of time, change of interests) only by 37 percent (in points: 16.9 out of 45.6). The remaining 63 percent of the spread of marks (in points: 28.7 out of 45.6) can be explained by two main reasons: the distance format of the entrance exam and partial distance learning in the first year.

In the presented study, the ratings of not only applicants based on the results of university entrance testing, but also those entering based on the results of the USE were analyzed.

The spread of scores in mathematics based on the results of the introductory testing conducted in the online format in 2020 and according to the results of the examination sessions, compared with the previous three years, when the introductory testing was conducted in the offline format, was 28.7 points. At the same time, a similar spread of grades for applicants based on the results of the USE, which all these years was held exclusively in the offline format, amounted to 7.7 points.

Therefore, 17 percent (in points: 7.7 out of 45.6) of the deviations in the marks of applicants in mathematics in 2020 in the results of online entrance tests and examination sessions can be explained by the distance (partly distance) learning format in the 1st year, and the online format of the entrance examination as the reason for the deviation in grades accounts for 46 percent (in points: 21 out of 45.6).

A detailed analysis of the factors influencing students' grades in mathematics upon admission based on the results of the entrance university online testing and on further education at the university based on the results of examination sessions showed the following. The correlation between math scores at university entrance and at university is positive, but only 0.22. This means that the mechanism of competitive selection of students for enrollment based on the results of university online testing does not make it possible to accurately predict student performance in the future.

The deviation in the scores of students before and after enrollment is mainly explained by the value of the scores based on the results of examination sessions (correlation 0.9). This indicates a low dispersion in marks in mathematics upon admission and insufficient differentiation in the level of knowledge of applicants. However, two ranges of student input ratings stand out strongly: 55–60 points and 90–95 points. Apparently, so many points were often scored by applicants who used outside help in the introductory online testing in mathematics. The purpose of the first category of such applicants was probably to recieve a minimum positive assessment, sufficient for enrollment in non-budgetary places or for enrollment in budgetary places in non-prestigious areas of training and specialties. The goal of the second category of such applicants is to achieve the highest possible score necessary to win the competition for budget places in prestigious areas of training and specialties.

The propensity of applicants to cheat on exams and turn to outsiders for help depends, among other things, on the age and level of previous education and belonging to the citizenship of a particular country (correlations with the entrance rating in mathematics are −0.18 and −0.15, with a deviation in student scores up to and after enrollment of 0.09 and 0.11).

The analysis of the correct answers in the online test revealed five tasks, the success of which most affects the model for assessing students' knowledge of mathematics. In connection with the foregoing, it may be quite appropriate to specifically add "trap tasks" to online tests, the correct answers to which should draw attention to the incoming from the moderator.

Entrance online testing to the university in 2020 has always been carried out using the webcams (cameras) of applicants and under the constant supervision of moderators; this makes it possible to expand the set of initial data on students for future research. Gross violations, such as strangers in the frame and conversations with someone, were identified immediately. However, each camera is known to have a limited viewing angle, and some applicants may have taken advantage of prohibited materials and outside help. Thus, this study is planned to be continued with the aim of automatically detecting violations based on images from webcams (cameras) of incoming ones as well as detecting suspicious patterns of behavior of the tested and issuing the corresponding warning. For example, the authors of works (Özgen et al., Li et al. [39,40]) on creating fraud detection mechanisms in online interviews and online exams suggest tracking the absence of a person in the frame, the presence of an outsider or another person in the frame, tracking the movements of the head and torso, and moving the computer mouse pointer across the screen. These ideas are also planned to be implemented as part of a future study.

**Author Contributions:** Conceptualization, A.S. and M.G.; Methodology, A.S. and M.G.; Software, M.G.; Formal analysis, M.G.; Investigation, M.G.; Resources, M.G.; Data curation, A.S.; Writing—original draft, A.S. and M.G.; Writing—review & editing, A.S.; Supervision, A.S.; Project administration, A.S.; Funding acquisition, A.S. All authors have read and agreed to the published version of the manuscript.

**Funding:** This research received no external funding.

**Institutional Review Board Statement:** This study was conducted using an anonymous online entrance testing and did not require ethical approval.

**Informed Consent Statement:** Informed consent was obtained from all subjects involved in the study.

**Data Availability Statement:** All dataset will be made available on request to the corresponding author's email with appropriate justification.

**Acknowledgments:** We are very grateful to the reviewers for their valuable comments, which helped to improve the article. We thank Pavel Cherdantsev, Executive Secretary of the AltSTU for support and assistance in data collection.

**Conflicts of Interest:** The authors declare no conflict of interest.

## Abbreviations

The list of abbreviations used in the manuscript:

| Abbreviation | Explanation of the Abbreviation |
| --- | --- |
| SEE | Sim Entrance Exam |
| UMAT | Undergraduate Medicine and Health Sciences Admission Test |
| ENEM | Exame Nacional do Ensino Médio |
| USE | Unified State Exam |
| RF | Russian Federation |
| AltSTU | Altai State Technical University named after I.I. Polzunova |
| VK | Vkontakte |
| SVE | Secondary vocational education |
| HE | Higher education |

## Appendix A

**Table A1.** Baseline data on admission and subsequent studies of 553 students enrolled in the university in 2020 based on the results of online entrance testing (for example, two students).

| Attribute | Value in Line 1 | Value in Line 2 |
| --- | --- | --- |
| Human test code | 1732 | 2332 |
| Test time, minutes | 144.5 | 88.6 |
| Admission math score (Based on online testing results) | 70 | 51 |
| The correctness of the applicant's answers for 25 tasks in the test in mathematics (primary scores [3]): | | |
| Tasks Nos. 1–5 | 1 0 0 1 1 | 0 1 0 0 1 |
| Tasks Nos. 6–10 | 0 1 0 1 2 | 1 0 0 1 2 |
| Tasks Nos. 11–15 | 0 0 0 2 0 | 0 1 0 0 0 |
| Tasks Nos. 16–20 | 0 1 0 0 1 | 0 1 0 0 1 |
| Tasks Nos. 21–25 | 1 1 0 1 0 | 1 0 0 0 0 |
| Incoming information: | | |
| Gender: M or F | M | M |
| Age (as of 11/01/2020) | 18 | 32 |
| Nationality, country | Kazakhstan | Russian Federation |
| University location region | - | yes |
| Settlement type: city or village | City | City |
| Data on previous education incoming: | | |
| Type of education document | General secondary education certificate | Diploma of Vocational Education |
| Year of education | 2020 | 2019 |
| Average score in mathematics in 2020 for applicants from educational institutions [1] (number of such applicants) | 80.5 (2) | 51 (1) |

**Table A1.** *Cont.*

| Attribute | Value in Line 1 | Value in Line 2 |
|---|---|---|
| University enrollment information:<br>Direction of preparation<br>Form of study<br>Financing conditions [2]<br>Average score in mathematics in 2020 enrolled in the same area of study and in the same form of study (number of enrolled) | Power Engineering<br>Full-time<br>Budget place<br><br>56.6 (47) | Power industry<br>Part-time<br>Off-budget place<br><br>77.42 (73) |
| Score in other subjects at admission | 46.5 | 48.5 |
| Score in mathematics when studying at the university (Based on the results of exam sessions)<br>Student status | 0<br><br>Expelled | 30<br><br>Continues education |
| Variety of grades in mathematics at university and at admission | −70 | −21 |

[1] An educational institution that an applicant graduated from before enrolling in a university (usually a secondary school or college). [2] Conditions for financing education: budgetary place—with payment of tuition fees from the federal budget, and non-budgetary place—with payment of tuition fees by an individual (student or his parents). [3] The maximum score for the correct answer in tasks Nos. 10, 14, 16, 23, 25 is 2.

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
