# Peer review of "Factor Analysis of Students’ Knowledge Assessment Based on the Results of Online Entrance Testing in Mathematics to the University under the Conditions of COVID-19"

_education, doi:10.3390/educsci13010046_

Round 1
Reviewer 1 Report
- -Very interesting. It is novel in its research
- -Encourage you to continue with further research on the subject.
- -Originality and innovation in the study
- -Congratulations on the references
Aspects that can be improved :
- Several citations from 2014, which contrasts with the many that are collected in the research that are current, since they have many from 2022
- There are many references in the middle of the sentence, when they should preferably go at the end of each one
- For example, in Figure 6, as in others, a lot of information is given in its title, when perhaps it should be in the legend.
- Figures should not have vertical stripes
- Reference nº10 has a point before the doi, instead of a comma like all the others.
- Point 4.3. It begins by referring to the table below, better if it has some previous phrase of an anticipatory nature
Author Response
Dear Reviewer,
Thank you for your interest in our work and valuable comments!
We attach our responses to the reviewers' comments as a separate document.
Best regards,
All authors

Reviewer 2 Report
The term MATHEMATICS should appear in the title of the article, as the factor analysis carried out is about students' knowledge of mathematics.
ABSTRACT: The abstract does not follow the IMR&D structure (INTRODUCTION, METHOD, RESULTS AND DISCUSSION). The abstract gives a lot of data on the results but does not contextualise them, nor does it present the reason for the study, nor explain the objectives and method used.
INTRODUCTION. It is well done, contextualises the study and presents other related research. It is interesting that, at the end of the introduction, the contributions of the work carried out are expressed.
THEORETICAL FRAMEWORK. Related works is confused with a large part of the Introduction. The Theoretical Framework should be arranged in a single section and the Introduction should be left to contextualise and set out the research problem.
The Theoretical Framework lacks references to European studies.
METHODS. The objective of this study appears at the end of the INTRODUCTION and is expressed as follows: The purpose of this study was to construct a factor analysis of the knowledge assessment of mathematics students enrolled in 2020 at the Altai State Technical University named after I.I. Polzunova (AltSTU) based on the results of the entrance tests conducted during the COVID-19 pandemic in an online format using an Internet browser and a webcam of the applicants.
But no research question appears before
In conclusion, a comparison is made between the results of other years compared to the results of students who entered university at the time of the pandemic. This should be made clear in the abstract and in the title of the article.
RESULTS. 3.1. Statistical data, refers to the characteristics of the sample. this should be clearer in the title of this section. A clarification should be made for the reader on how the results are presented in the different sections that appear in section 4.
DISCUSSION CONCLUSIONS: He argues that the deviations detected in the results are due to personal problems of the applicant, difficulties in adapting to university, lack of time, change of interests. How does he reach these conclusions if there is no data to support these claims? None of these variables have been studied
In another part of the discussion it is stated: The propensity of applicants to cheat in exams and to ask for help from outsiders depends, among other things, on age and previous education level, citizenship of a given country (the correlations with the access grade in mathematics are -0.18 and -0.15, with a deviation in the students' grades up to and after enrolment of 0.09 and 0.11).
How do you arrive at this statement? How can you say that the students cheated? Has a student-by-student investigation been carried out to find out if they cheated? Or is this a conclusion that the authors reach because there is a deviation from the access grades to the first year grades? I think these are assertions that can be made if no concrete data is provided. This part should be reviewed and in any case data should be provided on what external aid they actually received. It is planned to continue the study by analysing the webcams to check whether the students cheated in the exams. In the meantime, the assertions made in this article cannot be made.
State more clearly what conclusions have been drawn from this comparative analysis and whether the objective(s) of the study have been achieved.
The limitations of the study should be described
Author Response

(The authors gave the same response as above.)

Reviewer 3 Report
This research reaffirms the problem found during the online entrance exams, objectively. Virtuality has come to stay and to get the most out of it, the results of this research will lead to finding a solution to the problems encountered.
Author Response

(The authors gave the same response as above.)

Round 2
Reviewer 2 Report
The changes made to the article are satisfactorily in line with the recommendations made.
It only remains to explain the limitations of the study.